# "…pretty much all white, and most of them are psychiatrists and men": Mixed-methods analysis of influence and challenges in global mental health

Farah Shiraz[1], Marianne Ravn Knop[2], Amanda Low[2]*, Anna Durrance-Bagale[3]*, Max D. López Toledano , Helena Legido-Quigley[3,4], Natasha Howard[2,3]

**1** Khora Virtual Reality, Copenhagen, Denmark, **2** Saw Swee Hock School of Public Health, National University of Singapore and National University Health System, Singapore, **3** London School of Hygiene & Tropical Medicine, Department of Global Health & Development, London, United Kingdom, **4** Imperial College London, London, United Kingdom

* alpt@nus.edu.sg (AL); Anna.Durrance-Bagale@lshtm.ac.uk (ADB)

## Abstract

Mental health is increasingly recognised as a global health priority, with 'global mental health' gaining relevance as a field. We thus aimed to identify the most influential actors in global mental health and key challenges in the design and implementation of policies and interventions within this relatively nascent field, to provide suggestions on how mental health could be promoted and diversified at the global level to improve mental health outcomes. We conducted a mixed-methods study, incorporating a social network analysis of 115 experts to identify the most influential mental health actors and elements globally and 30 semi-structured key informant interviews, analysed thematically, to examine key challenges in developing the field. We found concentrated influence among a few actors, with network analysis highlighting psychiatry followed by psychology as most influential specialties, and academia as the most influential sector; limited global collaboration and political engagement; and the need for greater professional, socio-cultural, geographical, and gender diversity. Sustaining mental health prioritisation globally requires directed and coordinated efforts from influential individual and institutional actors in the global mental health field. This requires critical engagement with dominant modes of knowledge production, greater synergy at global, national, and community levels, and agenda setting by a broader and more equitable coalition of global mental health actors across professional, cultural, and gender differences.

## Introduction

Mental health is increasingly recognised as a critical aspect of population health and wellbeing globally [1,2]. It was designated a health priority in the 2015 Sustainable Development Goals (SDGs) Agenda 2030 and recognised as a fifth non-communicable disease in the World Health Organization Mental Health Action Plan 2013–2020 [3,4]. As many countries worldwide increasingly share a greater number of mental health concerns, the field of global mental

**Data availability statement:** We have made raw data for the Social Network Analysis available on a public repository platform. i. The data is located on the platform, Open Science Framework. ii. The link and DOI of the data set is as follows: https://osf.io/rs7ed/ and DOI 10.17605/OSF.IO/RS7ED

**Funding:** The Singapore Population Health Research Impact Centre (SPHERiC) provided funding for data collection and analysis (reference 026), through an award to FS and NH, but had no role in study design, data collection, analysis, or interpretation. Thus, results and interpretations do not necessarily reflect funder views.

**Competing interests:** The authors have declared that no competing interests exist.

health developed over the last two decades [5]. Early impetus was provided by a Lancet series published in 2007 arguing for urgent recognition of mental health as a global health priority, using the slogan "no health without mental health" [6–8]. Despite this, other areas of global health continue to receive greater prioritisation and funding [1,2]. While the neoliberal "deep core" of contemporary global health governance and policymaking frames global health issues as problems to be solved through cost-effective allocations of finite resources [9,10], even relatively "unfettered healthcare markets" are heavily influenced by factors such as political interests, security concerns, or involvement of influential actors [11,12]. Thus, global health fields that effectively advance a vision aligning with such factors (e.g., global health security) are generally prioritised [3,10].

In the case of mental health, this was made apparent during the COVID-19 pandemic. Pandemic consequences that affected mental health ranged from ruptures in day-to-day lives to destabilisation in international and national systems of trade, operations, and migration flows and exacerbation of widespread structural inequalities [13–16]. The World Health Organisation (WHO) highlighted interruptions to critical mental health services in 93% of countries, which jeopardised provision in an already underfunded field – especially in low and middle-income countries (LMICs) [6–8, 17–19]. With the pandemic perceived globally as a major economic and security threat, one effect was the heightened prioritisation of mental health as a global issue and increased attention to the field [17–20]. Whether this will last is unclear. It is therefore crucial to identify actors with the perceived authority and legitimacy to influence decision-making for the mental health field at the global level, as post-pandemic attention to mental health wanes. Existing attention needs to be properly coordinated towards reorganising mental healthcare systems and developing and implementing policies and interventions that improve psychosocial outcomes, so that this increased global prominence, even if temporary, has value [21].

Historically, addressing the mental health of populations focused on investigating severe mental disorders in Europe and North America, with some cross-cultural explorations but limited "global" thinking [5]. Separate but related developments in academia and practice converged to inform the beginnings of the contemporary field of global mental health. Some of these developments included growing evidence of systematic injustices experienced by people with mental health conditions across the world and studies that demonstrated a range of effective treatments for mental disorders in low and middle-income countries (LMICs) [5]. Global mental health can thus be considered a sub-field of global health involving consolidation of the efforts of mental health actors across international borders, sectors, and professional specialties to improve the mental health of populations globally [22–24].

Global mental health, as a field of theory and practice within the "global" space is processual and requires the "historically situated, distributed work of a multitude of actors" [25]. The construction and legitimacy of global mental health and prioritisation of mental health amidst other global health concerns thus requires continuous iterative action from its influential actors to demonstrate its value and potential directions for the field. Against a backdrop of competing global health priorities, there is an urgent need to identify influential actors in this nascent field, to determine challenges, and to ascertain possible directions for its future development.

We thus conducted a reanalysis of data collected during the COVID-19 pandemic, aiming to identify the most influential institutional and individual actors in global mental health and key challenges to its global prioritisation. Objectives were to: (i) conduct influence mapping of the field; (ii) identify perspectives of those working in the field on its key influences and challenges; and (iii) provide recommendations on actions and relevant additional actors to advance the field and improve mental health outcomes.

## Methods

### Study design

We adopted a sequential explanatory mixed-methods design, including data from social network analysis survey and semi-structured key informant interviews collected December 2019 to October 2020. Adopting this approach enabled us to identify influential individual and institutional actors in this field and explore the processes and relations constituting this work. First, narrative literature and document reviews of articles and technical reports on global mental health policy and prioritisation enabled development of survey questions, interview topic guides, and an initial seed list of potential participants [3]. Second, social network analysis (SNA) mapped and measured social relations between actors in a network, enabling interpretation of complex relational data on the structure and form of social relations viewed from an "outsider perspective" [26]. Third, explanatory qualitative interviews enabled deeper contextual understanding of SNA results, helping clarify the content and processes of how social relations were formed, maintained, and negotiated [26].

Our research question was: "Who are the most influential individual and institutional actors in global mental health, what are key challenges in the design and implementation of global mental health policies and interventions, and how can the field be promoted and diversified within the broader global health context?"

### Data collection

**Social network survey.** We compiled a seed list of 28 global mental health actors from two global mental health reports, namely the Lancet Commission on global mental health and sustainable development [3] and the ASEAN Mental Health Systems report [27]. We sent email invitations between December 2019 and December 2020, including up to three follow-up contacts. Those providing written informed consent via email were asked to complete 4 open-ended questions (i.e., who do you think are the ten most influential people in global mental health; who are the ten most influential institutions in global mental health; who do you regularly work with on mental health issues (people or organisations); whose views would you recommend we seek on global mental health?) We then used snowballing, by contacting those nominated and repeating the process until we achieved saturation of nominated actors. This approach allowed us to identify actors in the field perceived as 'most influential' and map interactions between them [28].

**Interviews.** We developed an interview guide, based on SNA findings, including experiences of global mental health work, prioritisation of mental health globally, changes in perceptions or issues needing to be addressed, challenges in prioritizing and implementing mental health policies globally, and anticipated future of mental health. We used maximum variation sampling of our network survey sample, by geographical region (i.e., Africa, Americas, Asia, Europe, Oceania), high-income or low/middle-income country (LMIC), gender (i.e., male or female, as none identified as non-binary), professional specialty, and sector to allow greater insight into perceived challenges and wider prioritisation issues. All provided verbal informed consent prior to participation. FS conducted interviews remotely in English via Zoom or Skype, each lasting approximately 50 (range 45–60) minutes and audio recorded, with field notes and reflexive journaling undertaken immediately after. We removed personally identifiable content from transcripts and assigned identification codes to help ensure anonymity.

### Analysis

MK conducted social network analysis of survey data in RStudio version 1.3.959 [29]. This entailed categorising participants and nominated actors into sectors and professional expertise

based on organisations they represented and education/job title provided, then converting actors (node-lists) and ties between actors (link-lists) into a graphical matrix of the network, and calculating common network indicators.

FS and AL analysed interview data inductively, conducting thematic analysis in NVivo software (Version 12) as described by Braun and Clarke [30]. This involved data familiarisation through reading/re-reading, generating initial codes, searching for themes, collating codes to generate themes, refining themes to ensure coherence codes within each theme and validity of each theme in relation to the dataset, and discussion and consensus with co-authors (MRK, HLQ, NH) on interpretation.

### Ethics

The National University of Singapore institutional review board provided ethics approval (S-19-339).

## Results

Table 1 provides characteristics of 115 survey and 30 interview participants. Most survey participants were male (55%) and interview participants were female (60%), most were from academia (15% and 47% respectively), followed by civil society (33% for both). Most were psychologists (32% and 27% respectively) followed by psychiatrists (25% and 23% respectively), and based in Europe (46% and 60% respectively).

**Table 1. Participant characteristics.**

| Characteristics | | Survey n=115 (%) | Interviews n=30 (%) |
|---|---|---|---|
| **Gender** | Female | 52 (45) | 18 (60) |
| | Male | 63 (55) | 12 (40) |
| **Sector** | Academia | 57 (50) | 14 (47) |
| | Civil society organisation | 38 (33) | 10 (33) |
| | UN agency | 11 (10) | 1 (3) |
| | Private sector | 4 (3) | 2 (7) |
| | Government | 3 (3) | 2 (7) |
| | Funder | 2 (2) | 1 (3) |
| **Specialty** | Psychology | 37 (32) | 8 (27) |
| | Psychiatry | 29 (25) | 7 (23) |
| | Epidemiology and public health | 24 (21) | 5 (17) |
| | Anthropology | 5 (4) | 2 (7) |
| | Neuroscience/biology | 5 (4) | 2 (7) |
| | Policy | 5 (4) | 1 (3) |
| | Human rights/law | 4 (3) | 2 (7) |
| | Sociology | 2 (2) | 1 (3) |
| | Lived experience advocates | 2 (2) | 2 (7) |
| | Nursing/Social work | 1 (1) | 0 |
| | Other | 1 (1) | 0 |
| **Region** | Europe [United Kingdom] | 18 [35] (46) | 7 [11] (60) |
| | Asia and Oceania | 26 (23) | 5 (17) |
| | Americas | 24 (21) | 4 (10) |
| | Africa | 12 (10) | 4 (13) |

### Network analysis of influential global mental health actors

Survey participants provided 1051 'most influential' nominations. By sector, most (700; 67%) nominations were for academics, followed by civil society and United Nations (134: 13% each), funders (32; 3%), government (24; 2%), private sector (18; 2%), and mass membership actors (9; 1%). By specialty, most (580; 55%) were for psychiatrists, followed by psychologists (229; 22%), anthropologists (98; 9%), sociologists (44; 4%), epidemiology/public health (27; 3%), lived experience advocates (23; 2%), neuroscientists/biologists (17; 1%), policymakers (12; 1%), lawyers/human rights advocates (10; 1%), nurses (5; 0%), and social workers/others (6; 0%).

Table 2 shows 256 actors nominated as having the most influence within the global mental health field. Each was nominated 1–93 times, with the 10 highest-ranked actors receiving 19–93 nominations. The most influential actors were predominantly male (55%) and in academia (60%), psychiatrists (38%) or psychologists (26%), and based in the US (57; 22%) and UK (56, 22%). Individuals working in funding, government, private sector, or membership organisations received less than 5% of total nominations, while similarly few individuals in neuroscience, nursing/social work, law, social sciences, policy, or lived experience advocacy were considered influential.

Fig 1 shows the network of influential global actors by speciality, with psychiatry and psychology dominating global mental health. Fig 2 shows academia as the most influential sector in global mental health. Clustering of UN and some CSO actors is notable due to the many ties between them, though this does not constitute a separate cluster as these are also closely tied to the academia-dominated network core.

**Table 2. Characteristics of 256 (116 female, 140 male) actors nominated as most influential in global mental health.**

| | | Female, n (%) | Male, n (%) | Total, n (%) |
|---|---|---|---|---|
| **Sector** (n = 256) | Academia | 59 (51) | 94 (67) | 153 (60) |
| | Civil society | 25 (22) | 19 (14) | 44 (17) |
| | UN agency | 9 (8) | 11 (8) | 20 (8) |
| | Government | 8 (7) | 8 (6) | 16 (6) |
| | Private sector | 8 (7) | 5 (4) | 13 (5) |
| | Funder | 6 (5) | 3 (2) | 9 (4) |
| | Mass membership | 1 (1) | 0 (0) | 1 (0) |
| **Specialty** (n = 256) | Psychiatry | 29 (25) | 67 (48) | 96 (38) |
| | Psychology | 37 (32) | 29 (21) | 66 (26) |
| | Epidemiology and public health | 14 (12) | 17 (12) | 31 (12) |
| | Other | 13 (11) | 7 (5) | 20 (8) |
| | Human rights/law | 4 (3) | 5 (4) | 9 (4) |
| | Neuroscience/biology | 4 (3) | 3 (2) | 7 (3) |
| | Anthropology | 3 (3) | 4 (3) | 7 (3) |
| | Policy | 6 (5) | 1 (1) | 7 (3) |
| | Lived experience advocates | 2 (2) | 2 (1) | 4 (2) |
| | Social work | 2 (2) | 2 (1) | 4 (2) |
| | Sociology | 1 (1) | 2 (1) | 3 (1) |
| | Nursing | 1 (1) | 1 (1) | 2 (1) |
| **Region** (n = 242) | Europe | 43 (39) | 57 (44) | 100 (41) |
| | Americas | 34 (34) | 35 (27) | 69 (29) |
| | Africa | 19 (17) | 17 (13) | 36 (15) |
| | Asia | 12 (11) | 11 (8) | 23 (10) |
| | Oceania | 3 (3) | 11 (8) | 14 (6) |

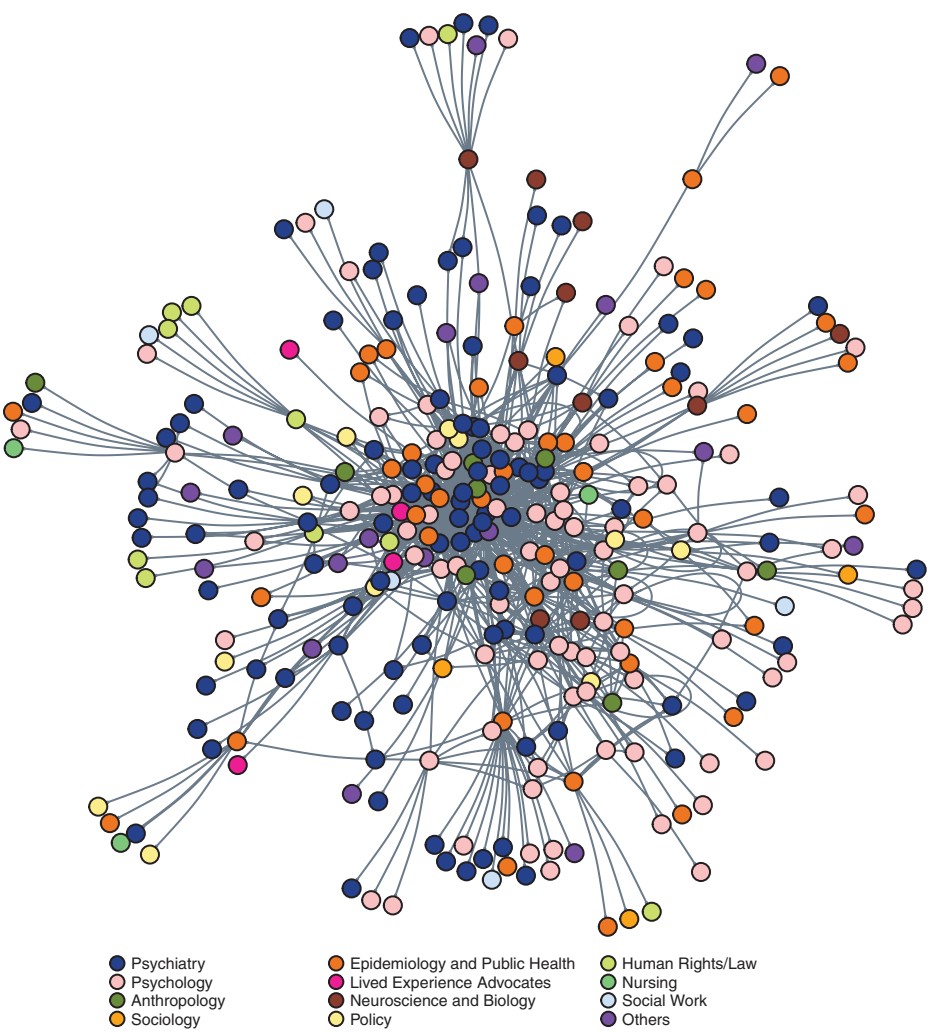

**Fig 1. Sociogram of most influential actors in global mental health by specialty.**

## Thematic analysis of interview data

Inductive analysis generated three themes: (i) overly concentrated influence; (ii) insufficient global collaboration; (iii) lessons for broadening involvement.

**Overly concentrated influence.** Participants discussed influence as insufficiently professionally, sectorally, geographically/linguistically, and gender diverse, remaining concentrated among male psychiatrists in 'Global North' academic institutions. Subthemes were thus: (i) dominance of academic psychiatry; (ii) dominant and disconnected geographies; and (iii) gendered influence.

**Dominance of academic psychiatry.** Participants described global mental health as 'psychiatry-led,' and the discipline was notably dominant within the field. Participants acknowledged that the origins of mental health as a practice, with its relation to biomedicine, meant that the historical focus on disease treatment and prevention within the field is understandable. However, other disciplines were largely neglected.

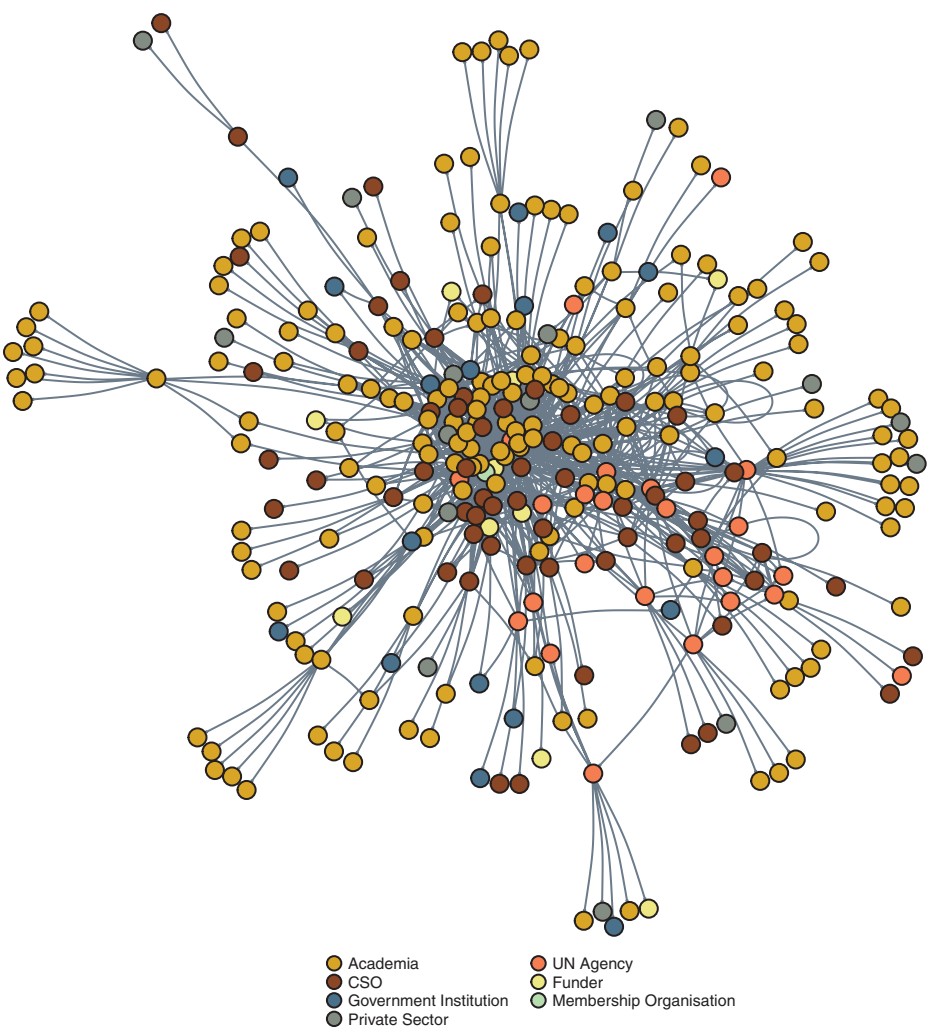

**Fig 2. Sociogram of most influential actors in global mental health by sector.**

"There is weighted or undue kind of prominence of psychiatry, and you see it really clearly everywhere in the world. The voices you hear most from are psychiatrists. I get very tired of people saying, "Oh, we only have two psychiatrists," and then I'm like, "Well, yes, there's X number of nurses. Social work, for example, I think is a hugely important profession within mental health and completely and utterly neglected in global mental health."

– Interviewee 030

The dominance of psychiatric perspectives related to the medicalisation of mental health. Participants suggested that if ministers only valued the opinions of mental health clinicians, missing relevant experience and knowledge from other professional disciplines (e.g., psychologists, anthropologists, nurses), they lacked crucial nuance necessary to reframe mental health globally.

While none disputed the need for clinical care of psychiatric disorders, mental health includes a spectrum of psychological and social experiences that relate to overall state of

mind and wellbeing. The field's reliance on psychiatrists and medicalised interventions came at the cost of limiting perspectives from professionals catering to myriad mental health and wellbeing needs of populations. Thus, many suggested the need for representation of holistic, preventative, and community-based mental health support as well.

> "I'd say [I work with] a lot of psychiatrists, especially a lot of our country research partners tend to be psychiatrists. In our NGO partnerships, it is a little more variable. A lot of the NGOs focus more on psychosocial interventions, [and] less of a medicalized mental health provider. They're more diverse in terms of their training and background."
>
> – Interviewee 026

**Dominant voices were also generally academic.** "I think that's where the challenge has been over the years because when you look at global mental health, it's been pushed primarily by academics and researchers from predominantly the Northern Hemisphere."

> – Interviewee 012

Whilst academics were accorded significant influence, criticisms were raised about how research from high to lower-income settings could be extractive and not necessarily translate into tangible benefits for research participants.

> "People come and people do a prevalence study…Then they walk away and nothing ever happens, you know? Or tons and tons of trials of interventions and they never get scaled up. So basically, nobody ever benefits from it."
>
> – Interviewee 007

**Dominant and disconnected geographies.** Expertise and influence remained embedded in high-income settings in Europe and North America, challenging meaningful alternative constructs of mental health. Some participants discussed what they termed the "colonisation of mental health," referring critically to this perceived global level disconnect between former colonial powers and former colonies in terms of what constituted mental health and relevant responses. Many questioned whether the 'global mental health' field could represent diverse settings given its metropole-led "one-size-fits-all" mental health constructs.

Another shortcoming of such disconnection became apparent during the COVID-19 pandemic, with some of the most powerful countries in Europe and North America struggled to contain the spread of SARS-Cov2 and address resulting mental health consequences. The ability of these countries to enact timely and appropriate policies to address urgent national needs and their corresponding global credibility was thus questioned. Previously regarded as models of mental health and policymaking, their influence may have weakened.

> "I think one thing COVID-19 has done is shaken up a bit of faith in the idea of the West as being the ultimate arbiters of knowledge. I think there's a sense of, "Well hold on, the US doesn't have its act together, to be sorting out everyone else"…"
>
> – Interviewee 022

An enduring legacy in formally colonised countries was the adoption of colonial languages, such as English or French. Mental health professionals are necessarily limited in the languages in which they worked, which effectively segmented the field linguistically.

> "Because of colonial histories, there is very little cooperation [between Francophone and Anglophone states] in relation to mental health. Global mental health is virtually absent in

the francophone African states. I went to a meeting of the World Psychiatric Association in Ethiopia and there was not one Francophone African representative there."

– Interviewee 030

Participants further highlighted ongoing disconnects between linguistic regions that operated largely independently with the 'global' limited to English speakers. For example, practitioners in Latin America typically worked with Spanish-speaking counterparts rather than in English, despite its dominance in global mental health. The global nature of the COVID-19 pandemic highlighted these disconnects.

"We didn't talk, I think at all, [about] South America. I haven't heard anything about other countries. Even East Europe, Russia or China. There is a problem of representation..."

–Interviewee 028

Thus, despite being labelled 'global,' this appeared somewhat aspirational in that the field was not equally relevant across global geographies and networks. Similarly, actions at the global supranational level may not necessarily synergise with initiatives at individual or community levels.

"Some of it might be people not choosing to be engaging globally [in] mental health because they don't think it serve their purposes you know. Certainly, that's true for some."

– Interviewee 002

**Gendered influences.**   Participants discussed how influential individuals and organisations skewed to male psychiatrists in Global North institutions.

"When you asked for the influential players in mental health, I'm like oh my god, they're like pretty much all white, and most of them are psychiatrists and men."

– Interviewee 007

Many participants highlighted that gender equity in mental health leadership could help counteract this insularity and ensure that diverse perspectives were included in collective decisions and lead to more positive change.

"A big challenge in mental health leadership globally is if you are a woman and non-English or American."

– Interviewee 009

Overall lack of diversity of influence meant equitable and engaged representation at the highest levels of decision-making remained unlikely as did consideration of mental health issues faced by underrepresented groups. However, most suggested that many within global mental health were open to more diverse involvement in leadership of the field.

"That's the other thing that needs to change in global mental health, not only do we need greater representation and diversity in terms of people of colour, but women. There's also a desire to see that change from a lot of people in the field."

– Interviewee 012

## Insufficient global collaboration

Participants consistently noted the limited global collaboration and coordination in mental health. For initiatives to start and develop sustainably, a group of actors able to dedicate time,

energy and resources is needed to drive initiatives until they gain traction. Involvement of small but committed groups has achieved milestones through cumulative efforts.

> "There are quite a few countries where through concerted effort of a small group of people that have been working with policymakers, we see shifts in policy. In all those efforts, there's a small group of people that have been working for years and years to get those changes."

– Interviewee 016

This dedicated core appeared lacking in many countries.

> "I think, one of the problems that we have in mental health, in a lot of countries, is you don't have a driver. Someone who drives the process. Someone who is passionate and prepared to make quite a lot of sacrifices, including financial sacrifices, to get certain things done."

– Interviewee 012

Several participants stated that dedicated professionals and resources were necessary but insufficient for mental health prioritisation in national agendas, as strategic leadership was required by international institutions to demonstrate global prioritisation. Some suggested that international bodies such as the World Health Organization (WHO) have not exercised their authority effectively to bring countries together to set a common agenda and drive its adoption in countries and regions. WHO's mandate means it operates primarily normatively and at the supranational level and is thus best at consensus-building rather than coercive authority.

> "What I see happening as the fallback option for governments is the WHO. And so, if the WHO reports a conservative way on [a matter] then at the end of the day you're not going to change the leadership… I see a fundamental role for WHO. But obviously there are all the NGOs and there are these regional initiatives. But it would be crucial for the WHO to really, at the regional level, take the lead then."

– Interviewee 025

Most participants mentioned ongoing efforts to improve mental health, whether at (sub) national, regional, or international levels. While such initiatives could be known to implementors, service-users, and policymakers by other terms than 'global mental health,' they shared similar agendas in pushing for mental healthcare development and reform.

> "Even before the whole movement around global mental health, people were doing their little bits to address mental health. But maybe we're not all doing it in the same direction and there has to be some consistency in terms of the way that mental health services are led and [ensure] equality in terms of access."

– Interviewee 021

> "I was not specifically looking for people with lived experience who are already involved at a global level. I was looking for people who might be [involved at] national [level] or even more locally involved; the idea [31] to empower them to get to the international level."

– Interviewee 005

While many participants acknowledged that the pandemic had detrimental and unequally distributed mental health consequences, it also increased global attention to the field. This propelled countries to come together with a greater sense of urgency and build upon existing efforts to constitute a more active response to shared mental health concerns across borders.

> "I really think that COVID was very positive in terms of putting some light on the mental health issue. I think that will really accelerate the war on mental health... Canada and the UK and Australia, they started to create this league of… mental health champions... It was really creating a new network of policymakers […] working together to promote mental health. And the first weighing in was two years ago in London in the UK. And I think quite a lot of countries were represented. Some by their own Ministry of Health and others by their delegates. It was a great success..."

– Interviewee 018

Beyond the temporary boost of heightened attention the pandemic gave to mental health, many suggested the global mental health movement was overly fragmented. For mental health to sustainably become a greater global priority in the mid- to long-term, greater clarity and consensus on objectives is required. While global bodies such as WHO needed to demonstrate leadership in prioritising mental health, national and regional actors needed to contribute and align with these initiatives for them to succeed.

> "You need to have one voice, regardless of the area or sector that you're working in, in global mental health. Whether you're in research, advocacy, or a user group, whatever is, we have to have one voice. That voice should carry the message and, from there, you create a critical mass of like-minded people."

– Interviewee 012

### Ensuring broader involvement

Participants advocated for greater involvement from under-represented professions, sectors, gender, and geographies to enable mental health to become a global priority.

> "We are [still] seeing influence in the traditional terms. I think traditional terms are - who has an impact on […] funding goals and actually getting that funding. That's majorly important on the content of what global mental health actually is… All of those are the more traditional parameters of determining influence and you get quickly to that group of people. In all fairness, that's not correct. If you're really looking at who are real global mental health champions, it's a health-worker in a clinic in rural Nepal who has influence and is actually doing a good job at integrating mental health at a primary health clinic..."

– Interviewee 016

Participants repeatedly identified specific actors (e.g., government policymakers, community organisations, service-users) as necessary in ensuring the sustainable success of mental health initiatives but were insufficiently involved at the global level.

**Government/policy engagement.** Many suggested politicians generally should be actively engaged, as their work afforded them access to decision-makers, contextual knowledge

of political priorities, and a position from which to initiate mental health policies and interventions at subnational and national levels, thus supporting the need for localisation and contextual adaptation of global initiatives.

> "How would a European researcher, or US-based researcher, get in good contact with or even understand the internal politics of any nation?... It needs someone on a national level who has that ability to step the bridge and say, 'I work with those, I'm not scared of those academic superstars… I'm not scared of them, and I can take their knowledge. I can discuss it, and I can bring it back and influence policy on that level'."
>
> – Interviewee 007

There are, however, barriers to greater involvement from political and other actors. Amongst these are political turnover, lack of policy continuity, and competing priorities that draw away attention and resources. Additionally, greater government discussion of mental health did not necessarily translate into tangible progress as political will and evidence to inform policy are also needed.

Participants described how the lack of political engagement hindering the progress of mental health initiatives can be partially attributed to a poor understanding of what mental health is. The COVID-19 pandemic provided an opportunity to explain mental health in more relatable and comprehensible ways to political elites. Whilst it is crucial for them to address concerns about mental disorders and conditions, taking a more holistic view of mental health enables opportunities to integrate mental health support in the day-to-day lives of the public and should thus rank higher on agendas.

> "… If we really want to put mental health [on] the agenda of policymakers... And if we really want to make mental health a priority, we need to educate people… [that] positive mental health is really a way of [tackling] the tension in your in your daily life. And that just managing your stress... and that would prevent not every major disorder, but still a large part. I think for me, in terms of lobbying, focusing on positive mental health would be more understandable by the population… Especially now with COVID-19. We're talking about dealing with emotions and dealing with stress."
>
> – Interviewee 018

Participants were unanimous that government actors alone could not ascertain mental health priorities and implement effective national initiatives, thus requiring collaboration with mental health stakeholders.

> "We work very closely with the head of mental health under the Ministry… [We have] this approach where the research funding and the grants that we get are not intended to provide any short-term funding that when withdrawn will just cause these programs to dry up. He works really collaboratively with the Ministry to determine what priorities are and how we should develop that. With the funding we get, how can we support evaluation or capacity building and things that we can invest in knowing that after those resources go away, you can continue on the work."
>
> – Interviewee 026

**Community-based organisations.**  Participants emphasised the importance of community-based organisations, especially their crucial role in identifying mental health needs in communities and ensuring the relevance and sustainability of interventions. External funders and implementer-led initiatives would not last without community support.

> "You regularly find that a lot of the experts who come in, are from overseas, come in and do their bit and then they leave. But there isn't that longevity."
>
> – Interviewee 021

> "I think there is a general understanding that services need to be community-based and I think they need to address issues identified by the communities, rather than us coming in and saying, 'This is a problem that needs treatment...'"
>
> – Interviewee 017

**Service-users/people with lived experience.**  Global mental health, as other fields, has increasing involvement of people with lived experience. Individuals who had or live with mental illness are consulted in the planning and implementation of some mental health initiatives. This works best when such perspectives are valued alongside those of mental health professionals.

> "The peer network often gets a request to review international documents with all of these guidelines. They then asked to review it and provide inputs. The peer network often also [gets requests] for lived experience participation at events. Let's say for example, there was an event on HIV and mental health. I get one of them who actually [has the] experience and skills in that area and they would go. One [event] was maternal mental health and [some]one would go to [that] event."
>
> – Interviewee 005

Several participants noted that involving those with lived experiences within global mental health advocacy required training and support for them to engage effectively with mental health professionals and to ensure their time and energy were not exploited or overused.

> "Those peer support groups, like many peer support groups, faltered because of the lack of funds and the reliance from people to run it themselves. Without any support, it's very difficult. People have enough struggles just surviving..."
>
> – Interviewee 030

If involvement of those with lived experiences is conceived and implemented strategically, their participation can underscore the salience of prioritising mental health in local and global agendas. Sharing experiences of living with and managing mental health conditions is typically expressed in ways that are most familiar to users. Although the unique voices of people with lived experience can highlight experiential realities and counter negative stereotypes, these perspectives may remain within certain social circles. This is especially true on social media platforms that allow for posting and sharing of user-generated content. For example, language can be either bridge or pose as a barrier when users post about their experiences as people primarily access and read content in languages they understand.

"At the moment, a lot of those voices are very present on social media. They're in English. I think it's quite concentrated within the urban middle class and the reach is fairly limited. Nonetheless, it's fantastic. I work with many of these people as well. It's fantastic to see people coming out openly on the media and talking about their experience."

– Interviewee 030

## Discussion

### Key findings and implications

This study is an initial effort to identify influence within the global mental health field, with qualitative findings adding depth to network analysis and indicating ways to advance the field beyond the biomedical model of global mental health explanation and knowledge production. Theorisation and leadership for global mental health as a field has come primarily from academic psychiatrists with our findings highlighting the benefits of expanding beyond academia and biomedical psychiatry [32,33]. Greater collaboration across sectors, professions, geographies, and gender will strengthen the field, help ensure more equitable leadership, and broaden the focus beyond clinical approaches [33].

The COVID-19 pandemic exposed the differential distribution of mental health outcomes amongst countries and populations and inadequacies in existing efforts to advance global mental health initiatives, demonstrating how mental health is continually affected by factors beyond health [17]. As such, including perspectives beyond a biomedical approach focused on disease and treatment requires encompassing prevention and community psycho-social care, policy, education and campaigns to inform gradual shifts in attitudes and reduction of stigma even though clinical diagnoses and treatments remain crucial [34]. There is a need for a renewed emphasis on approaches that addresses the needs of affected individuals and their families, acting against known drivers of poor mental health [35]. One way to facilitate this shift is to conceive of holistic mental health approaches that encompass emotional, psychological and social wellbeing. This requires non-health professional involvement in planning and implementation [36], with both health and non-health professionals equipped with knowledge and skillsets to work effectively in multidisciplinary teams [34]. In the two decades since the term 'global mental health' gained traction, there has been progress in the involvement of additional professions and sectors [22]. Expanded involvement is increasingly commonplace albeit not without some resistance from those accustomed to or privileged by current norms [37–40]. Such implicit 'rules' or preferences were evident in some interviewee experiences working in the field, particularly in academia and research contexts.

The perceived influence of psychiatry and related preferences for biomedical knowledge generation and explanation within "policy-scientific" spheres imparts a circularity and exclusivity to knowledge production and sharing within the field [41]. This links to individualist approaches wherein "societal problems are seen to originate in individuals" and "healing is based on ameliorating individual suffering rather than changing structural relations of domination and subordination" [42]. Despite literature evidence demonstrating the range of psychiatric approaches to mental health beyond those informed by the biomedical model, these alternatives are not always highlighted in mental health research, planning, and practice [43–46] and this gap between knowledge and practice was reflected by interviewees. Considered alongside the historical roots of the global mental health field, further critical engagement of and within the field is needed [5,32]. Critical perspectives, from disciplines such as anthropology and sociology, could provide constructive avenues to renew ways of conducting knowledge generation and transfer to policy and practice to improve mental health outcomes [32].

As many interviewees highlighted, actively involving service-users and people with lived experience, by incorporating their perspectives into planning and implementation can complement existing ways of working within global mental health. However, associated challenges require that this is done with due care to mitigate the risks of misinterpretation and exploitation or further traumatisation of people with lived experience, and ensure adequate training and preparation of lay people to productively engage in professional mental health settings at the global level [47].

Professing to be "global" requires confronting what "global" constitutes of in terms of which countries are involved and whether participation/non-participation of certain geographies and sectors is troubling, especially considering colonial histories. Several participants noted that knowledge exchange and practice were confined to certain geographies and bounded by language, which was primarily English. Notably, the COVID-19 pandemic demonstrated that worldviews and approaches exported from the Global North may not translate into effective mental health policies and interventions elsewhere, thereby contesting its epistemic authority [17]. We need to interrogate what it means to adopt a "global" mental health agenda and what this entails at the supranational as well as national and subnational settings. Distinction between *global* and *local* can be either a chasm that divides or a duality that allows actors within the field to find alignment and commensuration in advancing mental health at different scales and arenas of action [48].

Several participants highlighted how COVID-19 provided an opportune moment for the global mental health field to further advocate for mental health to national governments and the wider global health community. COVID-19 made more tangible a subject that is stigmatised and often perceived to be abstract or irrelevant [49]. There is thus a need for greater supranational coordination, and greater inclusion of subnational and non-mainstream perspectives, to articulate a shared global mental health agenda. Developing such an agenda of relevance at national and subnational levels requires both broader engagement and clear leadership. Most participants listed at least one global body, such as WHO, as a key actor within global mental health, substantiating their views by indicating these international organisations are not leveraging their positions sufficiently to spearhead mental health strategy and agenda-setting. WHO, though constitutionally constrained to normative, directing/coordinating, and research/technical cooperation functions, remains well-placed to set the tone and direction for a global mental health agenda [50,51].

As global mental health is conceived and practised across a wide range of populations, context-specific approaches to mental health must be supported. Fields such as cross-cultural psychiatry or ethnopsychology highlight the cultural variability of symptoms previously thought to be universally expressed and emphasise the importance of mental health research to be contextually attuned [52,53]. Interviewees underscored the vital role that community-based organisations play in needs assessment and ensuring the feasibility and sustainability of mental health initiatives. Additionally, onus should be on national governments, where possible, to coordinate between international bodies and those at subnational levels. Many noted the success of policies and interventions was influenced by political championing and governmental commitment.

Emphasising the "global" in global mental health, through diffusion of power from currently influential actors to more peripheral stakeholders, could help advance the field. Constructing the legitimacy of global mental health requires continuous iterative action from actors within the field to demonstrate its value, while global coordination and "universality" require support for prioritisation and standardisation processes from influential individual and institutional actors [54]. Processes such as policymaking and implementation requires involvement of academic-psychiatric influencers alongside those identified as having 'novel'

influence or experiences, including those with lived experience of mental health conditions. COVID-19 showed the possibility of solidarity at global, national, and subnational levels and the importance of international knowledge diffusion and translation [55]. Renewed attention to mental health and momentum amongst global health actors in response to the COVID-19 pandemic should be harnessed to further the prioritisation of mental health at the global level. Challenges in sustaining and translating this momentum into strategy and formalised action that are cognizant of the unique issues across geographic settings must also be addressed [56]. The future of mental health should therefore continue to encourage adoption of different perspectives, contextually attuned implementation, and reciprocal learning.

## Limitations

Several limitations should be considered. First, we aimed to sample a diverse group of professionals working in the global mental health field. However, the initial influential actors identified were primarily English-speaking academics as our seed list was drawn from a Lancet commission report. To partially counteract this, we used snowballing to reach additional global mental health actors. However, this demonstrated the restricted global influence of non-English speaking actors. Acknowledging that the representation of our study participants is limited despite our efforts in conducting purposive sampling was an important self-reflexive step for us as scholars grounded in adopting critical approaches. Further research is warranted to address this limitation. Second, the individuals and institutions nominated in the questionnaire were dependent on how questionnaire participants interpreted the word 'influence.' We acknowledge that this presents only a homogenous view of influential global mental health actors and may not account for other existing networks of influential actors beyond the nominations from the respondents in the current sample. Third, our study was conceived broadly to ask global mental health actors who they considered to be influential in global mental health and about their experiences working within the field. Even though our representation of global mental health actors is limited, we obtained responses from participants who worked across many settings. As such, we have refrained from providing very specific suggestions and recommend further research to inform country-specific or community-specific solutions or to guide supranational efforts. Lastly, our reanalysis of data collected in 2019–2020, while still very relevant, should be interpreted with an awareness of the context and period.

## Conclusions

While the global mental health field remains fragmented, there is much potential and many opportunities to (re)prioritise and advance the field. This study identified influential actors and institutions in global mental health and investigated their perceptions of and experiences working in the field. To build and sustain the efforts of these mental health actors, we thus propose greater inclusion of preventive initiatives and more diverse collaboration across sectors, professions, geographies, and gender to strengthen the field, help ensure more equitable leadership, and improve mental health understanding and support provision globally.

## Author contributions

**Conceptualization:** Farah Shiraz, Marianne Ravn Knop, Helena Legido-Quigley.

**Data curation:** Farah Shiraz, Marianne Ravn Knop, Amanda Low, Natasha Howard.

**Formal analysis:** Farah Shiraz, Marianne Ravn Knop, Amanda Low, Anna Durrance-Bagale, Max D. López Toledano, Natasha Howard.

**Funding acquisition:** Farah Shiraz, Helena Legido-Quigley.

**Investigation:** Farah Shiraz, Marianne Ravn Knop.

**Methodology:** Farah Shiraz, Marianne Ravn Knop, Natasha Howard.

**Writing – original draft:** Farah Shiraz, Amanda Low.

**Writing – review & editing:** Farah Shiraz, Marianne Ravn Knop, Amanda Low, Anna Durrance-Bagale, Max D. López Toledano, Helena Legido-Quigley, Natasha Howard.

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
