## [Decision Letter · Decision Letter 0]

25 Sep 2024

PGPH-D-24-01857

“…pretty much all white, and most of them are psychiatrists and men”: mixed-methods analysis of influence and challenges in global mental health

Dear Dr. Durrance-Bagale

Thank you for submitting your manuscript to PLOS Global Public Health. After careful consideration, we feel that it has merit but does not fully meet PLOS Global Public Health’s publication criteria as it currently stands. Therefore, we invite you to submit a revised version of the manuscript that addresses the points raised during the review process.

Please attend to the reviewers' comments in full when revising the manuscript.

As noted by Reviewer 2, there is currently a lack of cohesion between the framing of the piece, the question it seeks to address, the method and the discussion. The piece thus requires a major revision if it is to be publishable. 

As noted by Reviewer 1, the piece makes reference to COVID-19 as an important period but then does not follow through on explaining the significance of this. 

Finally, as noted by Reviewer 1, the authors should avoid the equation of psychiatry with a narrow biomedical approach given that within this field there is a breadth of perspectives. 

Please ensure that your decision is justified on PLOS Global Public Health’s publication criteria  and not, for example, on novelty or perceived impact.

Please submit your revised manuscript by**21 October** . If you will need more time than this to complete your revisions, please reply to this message or contact the journal office at globalpubhealth@plos.org. Please include the following items when submitting your revised manuscript:

We look forward to receiving your revised manuscript.

Kind regards,

Michelle Pentecost

Academic Editor

Journal Requirements:

1. Your current Financial Disclosure does not have Grant Recipient. However, your funding information on the submission form indicates a Grant Recipient "Dr Farah Shiraz". Please indicate by return email the full and correct funding information for your study and confirm the order in which funding contributions should appear. Please be sure to indicate whether the funders played any role in the study design, data collection and analysis, decision to publish, or preparation of the manuscript.

2. We note that your Data Availability Statement is currently as follows: "All relevant data are presented in this paper."

3. Please upload a copy of Figure 1a and 1b which you refer to in your text on page 6 and 22. Or, if the figure is no longer to be included as part of the submission please remove all reference to it within the text.

Please provide separate figure files in .tif or .eps format.

Reviewers' comments:

Reviewer's Responses to Questions

**Comments to the Author**

1. Does this manuscript meet PLOS Global Public Health’s publication criteria ? Is the manuscript technically sound, and do the data support the conclusions? The manuscript must describe methodologically and ethically rigorous research with conclusions that are appropriately drawn based on the data presented.

Reviewer #1: Yes

Reviewer #2: Yes

2. Has the statistical analysis been performed appropriately and rigorously?

Reviewer #1: Yes

Reviewer #2: N/A

3. Have the authors made all data underlying the findings in their manuscript fully available (please refer to the Data Availability Statement at the start of the manuscript PDF file)?

Reviewer #1: Yes

Reviewer #2: Yes

4. Is the manuscript presented in an intelligible fashion and written in standard English?

Reviewer #1: Yes

Reviewer #2: Yes

5. Review Comments to the Author

Reviewer #1: This was an interesting paper in its attempt to answer some broad scoped questions.

While the authors presented complex findings a coherent and methodical manner, with an awareness of the limitations of the sample itself, there are some missed opportunities to interrogate more comprehensively the results that are presented.

A few general suggestions that would help improve the critical assessment of the results in the discussion would be as follows:

1. The paper goes to some lengths to differentiate to differentiate the field of “global” mental health as incorporating for eg social determinants of mental health more centrally than global health as a field. It feels like an arbitrary distinction given that the authors then throw in concepts of “public health” alongside “colonial” constructs of health. It will help to discuss this more critically – as currently it doesn’t really make it clear where and if the distinction lies, what the difference and value would be in conceptualising the field.

2. Similarly to above the idea that the field of “psychiatry” being equated with a biomedical model of mental health management feeds into an archaic (and false) narrative that the psychiatrists are only representative of a Biologically focussed approach. It will be helpful to give the reader a sense that this is not necessarily the case, and that perhaps the academic and non-clinical weight of those considered influential are not necessarily the norm.(The field of psychiatry itself has many subspecialties which are not biologically focussed at all) – and contributes to the stigmatization of mental health conceptualization. In fact, in the majority world there is more of a collaborative approach to mental health management policy and advocacy alongside psychiatrists/allied health and lived experience patient researchers. The paper will benefit from a more inclusive and thoughtful perspective of this.

3. The discussion would benefit from a bolder and more nuanced critique of the responses – acknowledging the narrow representation of the participants as has been highlighted. How has this centered the need/urgency for more specific focussed approaches rather than the general suggestions given in the discussion/concluding paragraphs?

4. As the authors introduce the impact and influence of COVID-19 pandemic early on in the paper as their point of departure, it was expected that there would be a more uniform thread on how this influenced the perception of who the influencers were? And the disparity highlighted between majority world and the the rest in terms of access, and response to the pandemic that invariably should have but didn’t quite center the value of the non-Western world in contributing to influence is an opportunity that the authors should explore and discuss in this paper to critically bring it all together.

Reviewer #2: My main problem with the draft is that there is an incongruence between the background (lack of relevance of global mental health despite its importance), the methods (identification of the most influential mental health actors and institutions through interviews and social network analysis), and the results (concentrated influence, lack of diversity). The piece needs a profound reorganisation to make the argument cohesive. As far as I see, the main contribution of the piece is to highlight the lack of diversity and representation in global mental health. However, if that's the main contribution, then the introduction needs to introduce that problem. At the moment, the introduction is very generic; it repeats what's already being said about the importance of mental health globally, but it doesn’t touch on the growing literature and discussion about diversity and representation in global health and global mental health. Focusing the entire piece on diversity and representation would better align the different sections of the piece.

Other specific issues:

Introduction: The first paragraph, before any context or results are given, is already making proposals about what should be done with global mental health. This section should only provide the context/situation, and proposals should be given once the findings are presented.

The second paragraph gives an intriguing rationale for the paper: "Against a backdrop of competing global health priorities and the “partial and temporary” nature of the work striving for universality within the field, there is a longstanding need to identify influential global mental health actors and determine challenges and gaps [4]. Doing so enables further areas of exploration and development to be delineated to help the field advance and attract greater prioritisation and resources." An immediate question is why?. In what sense does the identification of influential actors responds or solves the lack of priority and fragmentary nature of global mental health? It might do, but the article doesn’t explain this, and it is crucial.

The section called “thematic analysis of interview data” points to three themes created through inductive analysis: (i) overly concentrated influence; (ii) insufficient global collaboration; (iii) lessons for broadening involvement. This is methodologically clear and solid and reveals what the paper is really about. Again, this is why the introduction needs to change.

Findings:

Most quotations from interviews are interesting and rich. However, some require more explanation. For instance, Page 10 includes one that mentions the need to include people with lived experience at different levels, but there’s no mention of this in the analysis.

Another quotation whose content is not contextualised or adequately used is the last one on page 13, where the dominance of English is mentioned by an interviewee.

Discussion:

The discussion section is probably the weakest at present. For the most part, it consists of generic, well-established, and often repeated recommendations for the development of public mental health systems in low- and middle-income countries (LMICs). These are only tangentially informed by the qualitative findings. The discussion should stem directly from the findings.

There are points made in the discussion that are not sustained by the findings and where no justification is given. For example: “Despite potential difficulties, it is through greater representation of diverse voices that stigmatising mental health discourses will change.”

The piece calls for unity and supranational coordination on the one hand, and local adaptability, diversification of voices, and inclusion of subnational and non-mainstream perspectives on the other. There’s a degree of contradiction or tension between these two calls that the piece doesn’t unpack or develop. Exploring this tension and its implications would make the piece stronger.

6. PLOS authors have the option to publish the peer review history of their article (what does this mean? ). If published, this will include your full peer review and any attached files.

---

## [Decision Letter · Decision Letter 1]

9 Dec 2024

PGPH-D-24-01857R1

“…pretty much all white, and most of them are psychiatrists and men”: mixed-methods analysis of influence and challenges in global mental health

Dear Dr. Durrance-Bugale

Thank you for submitting your manuscript to PLOS Global Public Health. After careful consideration, we feel that it has merit but does not fully meet PLOS Global Public Health’s publication criteria as it currently stands. Therefore, we invite you to submit a revised version of the manuscript that addresses the points raised during the review process.

The reviewers note the improvements made to the manuscript, but still have concerns that need to be addressed before this paper can proceed. Please attend to all of the comments by both reviewers, specifically in relation to a clearer structure of the paper, threading the argument more clearly from introduction through to conclusion, and to providing a more fulsome account of the role of COVID-19. Please ensure that you effectively link the background, research question, analysis and result.  

We look forward to receiving your revised manuscript.

Kind regards,

Michelle Pentecost

Academic Editor

Journal Requirements:

Additional Editor Comments (if provided):

Reviewers' comments:

Reviewer's Responses to Questions

**Comments to the Author**

1. If the authors have adequately addressed your comments raised in a previous round of review and you feel that this manuscript is now acceptable for publication, you may indicate that here to bypass the “Comments to the Author” section, enter your conflict of interest statement in the “Confidential to Editor” section, and submit your "Accept" recommendation.

Reviewer #1:

Reviewer #2: 

2. Does this manuscript meet PLOS Global Public Health’s publication criteria ? Is the manuscript technically sound, and do the data support the conclusions? The manuscript must describe methodologically and ethically rigorous research with conclusions that are appropriately drawn based on the data presented.

Reviewer #1: Partly

Reviewer #2: Yes

3. Has the statistical analysis been performed appropriately and rigorously?

Reviewer #1: Yes

Reviewer #2: N/A

4. Have the authors made all data underlying the findings in their manuscript fully available (please refer to the Data Availability Statement at the start of the manuscript PDF file)?

Reviewer #1: Yes

Reviewer #2: Yes

5. Is the manuscript presented in an intelligible fashion and written in standard English?

Reviewer #1: Yes

Reviewer #2: Yes

6. Review Comments to the Author

Reviewer #1: The main comment I have is on is the COVID-19 inference. As mentioned in the original review – there was an expectation that seeing as this study data was collected during the pandemic, there would be some reflection on this? It is such an integral flash point in the understanding of the disparities in the global health space, and mental health itself was detrimentally affected. The paper currently doesn’t address the original comment/suggestion on how to integrate the contribution of COVID 19, but especially since the authors mention it – there must be some expansion and critique on this.

My original comment is as reference.

Additionally the authors now only mention that the data was collected in COVID in the last paragraph of the introduction, and they also mention only once in the results a comment by a participant.

If there is an intention to include a COVID reference then there must be some introductory thoughts around it, and more focus on its contribution in the discussion.

Reviewer #2: First, thank you for the effort you have put into revising this article. It is much improved, with greater clarity, and addresses many of my initial concerns. However, there is still room for improvement.

In both the introduction and abstract, the formulation of the research question could be made stronger and more explicit. A stronger research question would clearly connect the background presented to the relevance of the problem and the specific insights offered by the paper.

Currently, the most immediate and relevant context for the problem is summarised as: "Against a backdrop of competing global health priorities, there is an urgent need to identify influential actors in this nascent field, to determine challenges, and to ascertain possible directions for its future development."

While this identifies a need, the background of competing priorities is not explained in sufficient detail, nor is it clear why this creates a problem. Furthermore, the link between competing global health priorities and the need to identify influential actors is not adequately articulated. It is also unclear why identifying these actors will naturally lead to a determination of key challenges or inform future directions for the field’s development. Strengthening this argument will better support the research objectives and clarify their relevance to the global mental health field.

7. PLOS authors have the option to publish the peer review history of their article (what does this mean? ). If published, this will include your full peer review and any attached files.

**Do you want your identity to be public for this peer review?** For information about this choice, including consent withdrawal, please see our Privacy Policy .

Reviewer #1: **Yes: ** ANUSHA LACHMAN

Reviewer #2: **Yes: ** Cristian Montenegro

---

## [Decision Letter · Decision Letter 2]

27 Jan 2025

“…pretty much all white, and most of them are psychiatrists and men”: mixed-methods analysis of influence and challenges in global mental health

PGPH-D-24-01857R2

Dear Dr Anna Durrance-Bagale

We are pleased to inform you that your manuscript '“…pretty much all white, and most of them are psychiatrists and men”: mixed-methods analysis of influence and challenges in global mental health' has been provisionally accepted for publication in PLOS Global Public Health.

Best regards,

Michelle Pentecost

Academic Editor

Reviewer Comments (if any, and for reference):

Reviewer's Responses to Questions

**Comments to the Author**

1. If the authors have adequately addressed your comments raised in a previous round of review and you feel that this manuscript is now acceptable for publication, you may indicate that here to bypass the “Comments to the Author” section, enter your conflict of interest statement in the “Confidential to Editor” section, and submit your "Accept" recommendation.

Reviewer #1: All comments have been addressed

Reviewer #2: All comments have been addressed

2. Does this manuscript meet PLOS Global Public Health’s publication criteria ? Is the manuscript technically sound, and do the data support the conclusions? The manuscript must describe methodologically and ethically rigorous research with conclusions that are appropriately drawn based on the data presented.

Reviewer #1: Yes

Reviewer #2: Yes

3. Has the statistical analysis been performed appropriately and rigorously?

Reviewer #1: Yes

Reviewer #2: N/A

4. Have the authors made all data underlying the findings in their manuscript fully available (please refer to the Data Availability Statement at the start of the manuscript PDF file)?

Reviewer #1: Yes

Reviewer #2: Yes

5. Is the manuscript presented in an intelligible fashion and written in standard English?

Reviewer #1: Yes

Reviewer #2: Yes

6. Review Comments to the Author

Reviewer #1: this version has adequately addressed my comments and reads well now in its entirety.

Reviewer #2: Thanks for patiently addressing the suggestions and comments. It is my view that the article is now ready for publication.

7. PLOS authors have the option to publish the peer review history of their article (what does this mean? ). If published, this will include your full peer review and any attached files.

**Do you want your identity to be public for this peer review?** For information about this choice, including consent withdrawal, please see our Privacy Policy .

Reviewer #1: **Yes: ** Dr Anusha Lachman

Reviewer #2: **Yes: ** Cristian Montenegro
